# Study of the Effect of Wild-Type and Transiently Expressing CXCR4 and IL-10 Mesenchymal Stromal Cells in a Mouse Model of Peritonitis

**DOI:** 10.3390/ijms25010520

**Published:** 2023-12-30

**Authors:** Soledad Garcia Gómez-Heras, Mariano Garcia-Arranz, Luz Vega-Clemente, Rocio Olivera-Salazar, Juan Felipe Vélez Pinto, María Fernández-García, Héctor Guadalajara, Rosa Yáñez, Damian Garcia-Olmo

**Affiliations:** 1Department of Basic Health Science, Faculty of Health Sciences, Rey Juan Carlos University, 28922 Alcorcón, Spain; 2New Therapy Laboratory, Health Research Institute Fundación Jiménez Díaz, 28033 Madrid, Spain; luz.vega@quironsalud.es (L.V.-C.); rocio.olivera@quironsalud.es (R.O.-S.); h.guadalajara@quironsalud.es (H.G.); damian.garcia@uam.es (D.G.-O.); 3Department of Surgery, Faculty of Medicine, Universidad Autónoma de Madrid, 28029 Madrid, Spain; felipevelezpinto@gmail.com; 4Biomedical Innovation Unit, Division of Hematopoietic Innovative Therapies, Centro de Investigaciones Energéticas Medioambientales y Tecnológicas (CIEMAT), Centro de Investigación Biomédica en Red de Enfermedades Raras (CIBERER), Instituto de Investigación Sanitaria Fundación Jiménez Díaz (IIS-FJD, UAM), 28040 Madrid, Spain; maria@kiji-tx.com (M.F.-G.); rosamaria.yanez@ciemat.es (R.Y.); 5Surgery Department, Fundación Jiménez Díaz University Hospital, 28033 Madrid, Spain

**Keywords:** mesenchymal stromal cells, sepsis, CXCR4, IL-10, stem cell therapy

## Abstract

Sepsis due to peritonitis is a process associated with an inflammatory state. Mesenchymal stromal cells (MSCs) modulate the immune system due to the paracrine factors released and may be a therapeutic alternative. Three treatment groups were developed in a murine model of peritonitis to verify the effect of human adipose mesenchymal stem cell (hASCs). Additionally, a temporary modification was carried out on them to improve their arrival in inflamed tissues (CXCR4), as well as their anti-inflammatory activity (IL-10). The capacity to reduce systemic inflammation was studied using a local application (peritoneal injection) as a treatment route. Comparisons involving the therapeutic effect of wild-type ASCs and ASCs transiently expressing CXCR4 and IL-10 were carried out with the aim of generating an improved anti-inflammatory response for sepsis in addition to standard antibiotic treatment. However, under the experimental conditions used in these studies, no differences were found between both groups with ASCs. The peritoneal administration of hASCs or genetically modified hASCs constitutes an efficient and safe therapy in our model of mouse peritonitis.

## 1. Introduction

Sepsis is a clinical condition with a high incidence rate. An uncontrolled infectious process is the cause of sepsis and involves a struggle between the infectious agent and the immune system, generating a systemic inflammatory state. Sepsis due to peritonitis is usually triggered after an anastomotic dehiscence following colorectal surgery [1] and is associated with morbidity and mortality [2,3].

Current treatment aims to control the abdominal focus causing peritonitis, primarily relying on surgery, broad-spectrum antibiotic therapy, and systemic support measures (oxygenation, intravascular volume control, and nutritional support). Despite these efforts, some patients experience a state of inflammatory systemic response syndrome that can become refractory to treatment, thereby leading to the death of these patients [4].

Intra-abdominal infectious pathology and its systemic effects are the subject of many assays, which aim to unify concepts, advance diagnostic and prognostic methods, and analyze the biomolecular bases of the pro and anti-inflammatory response. These efforts may allow for the use of modern treatments. However, morbidity and mortality remain unacceptably high, especially in severe infections with organ dysfunctions. Given all the previous aforementioned reasons, it is of great importance for the clinician to not only establish a timely and appropriate diagnostic and therapeutic sequence in patients with this pathology but also explore new alternatives that may contribute to a decrease in the mortality associated with sepsis [5,6].

Mesenchymal stromal cells (MSCs) are a subpopulation of multipotent cells that can be isolated from various adult tissues [7,8]. The infusion of adipose-derived stem cells (ASCs) has been described as a therapeutic strategy for the treatment of diseases related to inflammation and tissue injury because they are potent modulators of the immune system and have the ability to regulate both the innate and adaptive immune responses [9,10,11,12,13]. Several studies have demonstrated that the protective role of these cells in sepsis can be mainly attributed to paracrine factors released by ASCs or other cells they interact with, such as interleukin-10 (IL-10) [14], prostaglandin E2 (PGE2), tumor necrosis factor-alpha (TNF-α)-stimulated gene/protein 6 or IL-6, among others [8]. In addition, other studies ascribe the therapeutic effect of MSCs in sepsis to the direct antibacterial activity mediated by the secretion of antimicrobial peptides such as cathelicidin Hcap-18/LL37, beta-defensin, and hepcidin [15].

In the last decade, some authors have proposed to improve the capacity of MSCs by promoting the expression of anchoring proteins such as CXCR4 or anti-inflammatory interleukins such as IL-10 [16,17]. In this study, the aim was to investigate whether such modifications are safe and could improve the condition of animals with abdominal sepsis compared to the treatment with unmodified MSCs. In particular, the therapeutic effect of human ASC (hASCs) therapy in an animal model of severe sepsis induced by cecal ligation and puncture (CLP) was assessed, which is the main model for polymicrobial human sepsis [18,19]. In the study, a single ASC dose of 1 million hASCs per animal was used, which is in line with previous articles that have shown that MSCs appear to be effective. Furthermore, the antimicrobial activity of hASCs was evaluated by analyzing cytokines and chemokines levels in blood and peritoneal lavage, as well as by histology analyses of different mice tissues. Moreover, a study was conducted to test the capacity of hASCs to reduce the systemic inflammatory state and organ dysfunction through the modulation of the immune response and expression of tissue protective/regenerative factors. Having said that, a comparison of the safety and therapeutic effect of wild-type hASCs and human mesenchymal stem cells (hMSCs) transiently expressing CXCR4 and IL-10 was made, aiming to generate an improved anti-inflammatory response for sepsis, in addition to standard antibiotic treatment.

## 2. Results

### 2.1. Cecal Ligation and Puncture Surgery (CLP)

All procedures were conducted by the same surgical team using the experimental protocol described in Materials and Methods. Surgeries were performed in batches of five animals. Within 24 h of the cecal ligation and puncture (CLP) surgery, 8% of the total number of animals died: two in the control group, two in the hASC-MOD group, and three in the hASCs group. Additional mouse groups were included to complete the proposed number of mice per group.

### 2.2. Assessment of Subjects after Surgery

All treated animals were monitored daily after surgery to assess each of the parameters indicated in Appendix A. In mice that did not receive cell therapy, a progressive decrease in weight was observed until the mice were euthanized. In marked contrast with this observation, hASCs-treated mice lost a maximum of 10% of their weight during the first 24 h post-treatment, which was later recovered. No significant differences were observed in the evolution of weight between mice treated with unmodified and modified hASCs (Figure 1).

Another relevant aspect during follow-up of the animals was the alteration of their behavior. Animals from the control group that did not receive hASCs treatment did not move and only made slight and clumsy movements in response to stimuli. Similar results were observed in the pain sign scale, despite daily analgesia. In contrast to this, mice treated with either group of hASCs only showed a slight difficulty to move for 2–3 days post-surgery.

Regarding mortality not associated with the surgical procedure, no significant differences were observed among the three groups of mice. Three animals died in the control group (two animals on day 2 and one on day 4), two animals died in the hASCs group (day 2), and one animal died in the hASCs-MOD group (day 4).

### 2.3. Comparative Assessment of Microorganisms in Various Medium

The microbiological study of the peritoneal lavage showed that after ligation and puncture of the cecum, the peritoneal fluid contained *Lactobacillusacidophillus*, *Actinomyces odontolyticus*, *Enterococcus faecalis and Staphylococcus aureus*, which grew on blood and chocolate agar. On anaerobic medium *Lactobacillus acidophillus*, *Bacteroides distasonis*, *Enterococcus faecalis and Propionibacterium acnes* were detected.

In peripheral blood, samples obtained 48 h after surgery did not show any growth of microorganisms in any of the plates, with the exception of the control group, where anaerobic colonies of *Lactobacillus acidophilus* and *Parabacteroides distasonis* were observed.

### 2.4. Perturbation in Inflammatory Markers to Treatment Conditions

To evaluate the systemic response to the different treatments, various cytokines and chemokines involved in immune, pro-inflammatory, and anti-inflammatory processes were analyzed in blood at different times (pre-surgery, 5 and 11 days post-surgery), and compared among the different groups of mice.

Cytokines and chemokines were grouped according to their functional properties. As observed in Table 1, analyzing the effect of surgery (control group) compared to the sham-treated group on day 5 post-surgery, an increase in all pro-inflammatory cytokines was observed, especially IL-1β, IL-12, IFN-γ, and TNF-α, and a decrease in IL-17 in mice treated by surgery. Regarding anti-inflammatory cytokines (IL-4 and IL-10), no significant variations were observed. There was also a significant increase in most regulatory cytokines, as well as in the analyzed chemokines, except for Eotaxin and RANTES, which were decreased.

When hASCs treatments were compared with the control group, we observed an increase in anti-inflammatory cytokines in both groups; there was an increase in IL-4 in the hASCs group and an increase in IL-10 in the hASCs-MOD group. Furthermore, a decrease was observed in IL-12, IL-9 and RANTES values, as well as an increase in MCP-1, IL6 and IL13 in both cell-treated groups. Finally, in the hASCs-MOD group, a decrease in GM-CSF and an increase in IL-5 were observed compared with the sham group. When analyzing the different treatment groups at day 11 day, no significant differences among the control and the two hASCs groups were observed.

### 2.5. Abdominal Tissue Damage

#### 2.5.1. Cellular Localization

First, the objective was to evaluate whether hASCs remained in the abdominal tissues after 5 and 11 days. Nevertheless, no consistent presence of hASCs could be observed in any case.

Then, an analysis of the histopathological signs of the immunomodulatory effect of hASCs and hASCs-MOD in different abdominal tissues at 5 and 11 days post-infusion was conducted.

#### 2.5.2. Quantification of the Inflammatory Infiltrate in the Cecum

All data about neutrophils, plasma cells and macrophages are included in Appendix A.

Neutrophils. In all study groups treated with hASCs, there was a decrease in this infiltrate, both at 5 days and 11 days post-treatment, which was more pronounced in the hASCs-MOD group (*p* < 0.05).

Plasma Cells. From day 5 to day 11, no increase in plasma cells was observed in hASCs groups, although this increase was striking in samples from the hASCs-MOD group (*p* < 0.05).

Macrophages. In both groups treated with cells, M2 macrophages were present at day 5 post-treatment and increased by the 11th day. The effect was more pronounced in hASCs compared to hASCs-MOD (*p* < 0.05) (Figure 2A).

#### 2.5.3. Tissue Damage in Intestinal Loop Walls

In the control group, the inflammatory infiltrate caused severe tissue damage in all layers of the cecum wall, the focal focus of the CLP lesion, both at 5 and 11 days of evolution. This damage was significantly less (*p* < 0.05) in the cell-treated groups (Figure 2B and Appendix A).

In the distal loops, the inflammatory infiltrate in each studied group is equivalent: in all groups, this infiltrate only alters the outermost layers (serosa and outer muscular). (Figure 2C,D).

#### 2.5.4. Histopathological Evaluation of the Spleen

To compare the therapeutic effect of ASCs, an analysis of the morphology of this organ during sepsis from an immune point of view was performed.

In the spleen of the control group, the periarterial lymphatic sheath (PALS) was very reactive, as compared to the treated groups where there was only a slight increase in the size of this zone (mild reaction, grade 1 as indicated in Appendix A). It should be noted that in the hASCs group after 11 days of evolution, the T lymphocyte zones remained small; it is the least reactive of all the groups (*p* < 0.05) (Figure 3A).

In the lymphocyte B-dependent area (marginal zone of the spleen), the most reactive group was the control group, where the sepsis situation was the worst. In the rest of the groups, there was almost no reaction (Appendix A.

#### 2.5.5. Macrophages (Figure 3B)

In both groups treated with hASCs, there was an increase in macrophages CD163 compared to the total number of macrophages present (CD68) both at day 5 and 11 after cell administration, during which the most striking effect was observed. No significant differences were observed between the cell treatment groups. In the control group, differences began to be observed after 11 days.

Taking into consideration all the results (follow-up, microbiology, cytokine and chemokine levels and histopathological evaluation) observed in mice peritoneal fluid, blood samples, and abdominal tissues, the infusion of hASCs (either modified or not) induced a significant decrease in the inflammation state and a more effective resolution of the septic process.

All thing considered, a statistically significant difference (*p* < 0.05) was observed between the follow-up and histological results of the groups treated with cells versus antibiotic treatment. We did not observe significant differences in any parameter between the groups with different cell treatments.

## 3. Discussion

Sepsis is a clinical problem characterized by severe systemic inflammation resulting from a dysregulated host response to infection. Within septic pathology, peritonitis is closely associated with anastomotic dehiscence, leading to fecal release into the peritoneal cavity. In recent years, attempts to design effective and specific therapies targeting components of the dysregulated host response have failed. Sepsis remains a clinical challenge that is expected to increase, especially given the emergence of antibiotic resistance. Therefore, there is a pressing need for novel therapies to address this clinical problem. Sepsis represents the early systemic inflammatory response [20,21], which, in its initial stages, is associated with a depletion of B and T lymphocytes and an increased rate of apoptosis in stromal cells [22,23].

The cecal ligation and puncture (CLP) model is presented as the gold standard experimental model in sepsis research [24,25]. Despite this, it is associated with significant variability in mortality due to factors such as the number of punctures, size of the needle, duration of cecal ligation, and above all, the length between cecal ligation and the end of the cecum. For this study, it was decided to follow the model proposed by Ruiz et al. 2016 [18], performing all cecal ligations at 1 cm using a single puncture. In our study, the 8% mortality associated with the surgical procedure within 24 h of CLP surgery is consistent with data previously reported [24].

Considering the immunomodulatory [26,27] and antimicrobial properties of MSCs [28,29], the intraperitoneal administration of adult human MSCs may constitute a novel therapeutic tool to treat sepsis [30,31,32]. Contemplating a potential clinical use of MSCs in peritonitis, it was decided to test the effect of a high dose of xenogeneic human ASCs to evaluate not only their efficacy but also the potential toxic effects of these cells in the proposed mouse model of peritonitis. The primary reasons for delivering human ASCs intraperitoneally were, on the one hand, the anti-microbial capacity described for human MSCs [28,29], and on the other hand, the possibility of preventing the retention of ASCs in the lungs, liver, or spleen, which is associated with the intravenous injection of these cells, thus favoring the interaction of ASCs with the inflammatory foci of the peritoneal cavity. Additionally, our study proposes the use of two types of ASCs, WT ASCs and ASCs transfected with CXCR4 and IL-10 bi-cistronic mRNAs, which previously showed a local anti-inflammatory response in mice [17].

Our results firstly indicate that no significant immune response was triggered by xenogenic ASCs, which is in accordance with the Hoogduijn and Lombardo hypothesis [33] that ASCs are immune-privileged cells. Although their mechanisms of action are still not fully understood, our results seem to indicate that xenogenic ASCs favor the resolution of inflammation and the inflammatory imbalance associated with peritoneal sepsis [34,35,36,37].

In this polymicrobial sepsis model induced by CLP, no significant differences were observed in the number of either type of ASC at 5 or 11 days post-administration in the peritoneal tissues. In terms of survival, both ASC groups markedly increased mice survival, though genetically modified ASCs always showed slight increases over the WT ASC group. The absence of a significant therapeutic effect of genetically modified ASCs with respect to WT ASCs could be due to the high therapeutic effect that these cells already have in this peritonitis mouse model, possibly combined with the short-lived effect of transfected mRNA.

The therapeutic effect of WT and genetically modified ASCs was associated with an overall reduction in the inflammatory response, specifically a reduced production of IL-12 and IL-17, in accordance with the previous results of Luo CJ et al. [38]. In this study, reduced levels of IL-17 and increased levels of anti-inflammatory cytokines (IL-4 and IL-10) were observed. The response observed in the spleen confirms the advantages of the peritoneal route as a system for the application of xenogeneic ASC treatment since in both ASC-treated groups, there was a mild reaction in the reactive areas of the spleen (LT and LB areas) in contrast with the control group, indicating that the intraperitoneal injection of xenogeneic ASCs constitutes a safe and effective treatment in peritoneal sepsis, according to CTKs and histological studies. Similar results with allogenic MSC have been observed by Tan Y et al. using an intravenous application of MSCs [32].

The host immune response is of vital importance for the onset, progression, and outcome of sepsis [22,38,39,40,41]. This response starts with the activation of proinflammatory molecules, as well as anti-inflammatory cytokines [42]. Previous studies demonstrated the potential of MSC therapy in elucidating the role of mediators secreted by MSCs, including keratinocyte growth factor and TNF-α-induced IL-6, in modulating the immune response to endotoxins (reducing TNF-α and macrophage inflammatory protein-2 and increasing IL-10 concentrations) and in promoting the restoration of immune balance. Released cytokines play a key role in regulating the host immune response. Multiple studies have demonstrated that MSCs decrease proinflammatory cytokine responses while increasing the concentrations of anti-inflammatory agents [28,31,32,33,34,35]. In our study, ASCs significantly reduced cytokine and chemokine concentrations (IL-1β, IL-6, chemokines and TNF-α) and improved survival after cecal ligation and puncture in mice. ASCs maintained their efficacy when administered 4–6 h after polymicrobial sepsis induced by cecal ligation and puncture in mice.

The analysis of the systemic inflammatory response 5 days after administration revealed that the plasma levels of proinflammatory cytokines (TNF-α, IFN-γ, IL1b IL-2, IL-3, IL-4, IL-5, IL-6, IL-9, IL-10, IL-12, and IL-13) in the control group were increased. In contrast, mice treated with ASCs exhibited a significant reduction in the proinflammatory cytokines, IL-9, IL-12, and IL-17, compared with the sham or control groups. When the anti-inflammatory cytokines (IL-4 and IL-10) were analyzed, high levels in the blood of mice treated with ASCs were detected. These results are similar to those described by other authors [16,17]. MSCs, as immunomodulatory cells, amplify the normal variation of inflammatory cells in response to an infection: they cause a greater decrease in neutrophils during the first days post-infection and a subsequent greater increase in plasma cells relative to the physiological situation without ASC treatment [43]. It has already been reported that MSC treatments cause an increase in the infiltrate of CD163+ macrophages [41], which is somewhat higher in the genetically modified ASC treatment group. These macrophages have an anti-inflammatory effect [43], which was shown to accelerate the resolution of the septic process.

The spleen is an organ of marked relevance during sepsis and systemic infection [44], aiding in the clearance of bacteria. The analysis of lymphocytes in the spleen of mice treated with WT ASCs or genetically modified ASCs showed an enhanced anti-inflammatory polarization of human T cells in mice treated with these cells compared with the control WT ASC group. The morphology of the T lymphocyte-dependent zone (PALS) and the B lymphocyte-dependent zone (ZM) during septic processes and in the presence of antigens in blood react and increase in size due to cell replication of LTs and/or LBs [44]. References from the control group showed that both splenic areas appear highly reactive as they correspond to high microbiological blood levels. It was found that in the ASCs and ASC-MOD study groups, the splenic reactivity was lower, with the immunomodulatory effect of ASCs being the most plausible reason. These morphological findings in the spleen corresponded with the histopathology observed in the cecum and intestinal loop of each group. Finally, tissue damage and inflammatory infiltrate in the layers of the cecum wall were greater in the control group compared with ASC group, once again with a very similar effect in both groups treated with ASC cells.

## 4. Methods

### 4.1. Isolation and Expansion of Adipose Derived Mesenchymal Stem Cells

hASCs were isolated and characterized according to the protocols described by Hervás-Salcedo R et al. [17]. Briefly, adipose tissue was obtained by liposuction from healthy donors after informed consent. The liposuction sample was washed with phosphate saline solution and subsequently disaggregated and digested with collagenase A (Serva, Heidelberg, Germany) at a final concentration of 2 mg/mL for 4 h at 37 °C with shaking. Afterward, the samples were filtered through 100 μm nylon filters (BD Bioscience, Franklin Lakes, NJ, USA) and centrifuged for 10 min. The cellular pellet was re-suspended in α-MEM (Gibco, Waltham, MA, USA) supplemented with 5% platelet lysate (Cook Medical, Limerick, Ireland), 1% penicillin/streptomycin (Gibco, Waltham, MA, USA), and 1 ng/mL human basic fibroblast growth factor (Peprotech, London, UK). Finally, cells were seeded at 10,000 cells/cm^2^ in culture flasks (Corning, MS, USA) and cultured at 37 °C and 5% CO_2_. During the expansion of hASCs, the medium was changed every 2–4 days, and adherent cells were serially passaged using 0.25% trypsin/EDTA (Sigma-Aldrich, Madrid, Spain) upon reaching near confluence (70–90%). For all studies, hASCs were used at passages from 4 to 8.

hASCs were immunophenotypically characterized by flow cytometry (Fortessa, BD Bioscience, Franklin Lakes, NJ, USA) using the following monoclonal anti-human antibodies: CD29, CD44, CD73, CD90, CD105, CD166, CD45, CD19, HLA-DR, CD14, and CD34. Data were analyzed with FlowJo version X (FlowJo LLC, Ashland, OR, USA). The osteogenic and adipogenic differentiation ability of hASCs were determined using the NH-OsteoDiff and NH-AdipoDiff Media (MiltenyiBiotec, Bergisch Gladbach, Germany), respectively, according to the manufacturer’s protocols.

### 4.2. Cell Modification

A codon-optimized sequence of human CXCR4 and IL-10 cDNAs (GeneScript, Piscataway, NJ, USA) was cloned into a pUC57 plasmid as bicistronic constructs using the E2A sequence according to Hervás-Salcedo et al. [13]. Additionally, an in vitro transcription of the construct was carried out using the 5X MegaScript T7 Kit (Ambion/Invitrogen/Thermo Fisher Scientific, Austin, TX, USA). A 3′-O-Me-m7G (5′)ppp(5′)G RNA Cap Structure Analog (ARCA; New England Biolabs, Ipswich, MA, USA) was added to the 5′-end, and a 3′-poly(A) tail was added using the Poly(A) Tailing Kit (Ambion/Invitrogen/Thermo Fisher Scientific, Austin, TX, USA). mRNA was synthesized and purified using the RNAeasy^®^ Plus Mini Kit (Qiagen, Hilden, Germany). To modify the hASCs, cells were cultured for 24 h, and then mRNA containing CXCR4 and IL-10 was mixed in Opti-MEM solution at a concentration of 1 µg/10^5^ cells. On the other hand, Lipofectamine Messenger (Invitrogen/Thermo Fisher Scientific, Waltham, MA, USA) was added to the same amount of Opti-MEM medium at a concentration of 5 µL/10^5^ cells. Both solutions were mixed and incubated for 5 min at room temperature. Then, the mixed solution was added to the cells for ASCs transfection. After transfecting, cells were used in less than 12 h.

### 4.3. Animals

All studies were approved by the animal care committee at University Hospital Fundación Jiménez Díaz in accordance with the Spanish Council of Animal Care guidelines (Ref. PROEX 084/16).

A total of 87 mice, 8-week-old female C57Bl/6J (Charles River, Lyon, France), were used, divided into 3 groups, with 2 follow-up times (5 and 11 days). Three mice were used as sham group without cecal ligation and puncture surgery (CLP) procedure. The 3 study groups were the following:

Control Group (CG). Twenty animals were subjected to CLP and antibiotic treatment (ATB) with enrofloxacin 10 mg/kg every 24 h for 3 days. Then, CLP was performed following the protocol described by Ruiz et al. in 2016 [15]. In this group, mice were divided in two time control groups, 5 (CG-5d) and 11 days (CG-11d).

Treatments groups.

-*hASCs Group*. This group of animals underwent CLP + ATB and 4 h after the intervention; they were injected with 1 × 10^6^ hASCs intraperitoneally. Twenty animals were sacrificed after 5 days (hASCs-Group 5d) and twenty were left to evolve until 11 days (hASCs-Group 11d).-*hASCs-MOD Group*. After CLP+ ATB, 1 × 10^6^ hASCs-MOD cells, human adipose derived stem cells transfected with a bicistronic mRNA carrying CXCR4 and IL-10, were administered intraperitoneally. There were also two subgroups according to time of evolution: 8 animals hASCs-MOD were sacrificed 5 days after administration and 8 animals hASCs-MOD were analyzed at day 11.

During surgery procedure, 8 animals died and were replaced to complete the groups.

### 4.4. Cecal Ligation and Puncture Protocol

Briefly, female mice were anesthetized with inhaled sevoflurane (Sevorane^®^, Abbott, Madrid, Spain) (chamber induction with 5% sevoflurane inspired fraction and O_2_ at 1.0 L/min and subsequent maintenance during surgery with 3.0% sevoflurane). Mice were positioned in dorsal recumbency, and the ventrums were shaved and disinfected with povidone iodine. Before starting surgery, 5 mg/kg of *adolonta* (HAUPT PHARMA LIVRON, Livron Sur Drome, France) was administered in all cases (analgesic 6thareapy). A ventral midline incision (1 cm) was made to allow for the exteriorization of the cecum. The cecum was ligated 1 cm from the apex with 3/0 silk suture and penetrated through-and-through with a 21-gauge needle. The abdominal incision was then closed in two layers with a 4/0 silk suture. Immediately after surgery, animals were fluid resuscitated and hydrated with 1 mL of saline solution injected subcutaneously.

### 4.5. Animal Follow-Up

On a daily basis, all animals were evaluated after surgery according to the monitoring protocol approved by the OEBA, mainly analyzing health, weight, mobility, and signs of pain (Appendix A).

At day 5 or 11, according to group and time, under inhalation anesthesia, 1 mL of peritoneal lavage and 2 mL of blood by cardiac puncture were obtained and spun down. Plasma was stored at −20 °C for further analysis. Afterward, animals were euthanized, and all tissues of the peritoneal cavity were collected and fixed in 10% formaldehyde at room temperature for histological study.

### 4.6. Bacterial Culture

In 11 animals of different groups and taking advantage of those who died during the trial and with a maximum of 1 h since their death, 1 mL samples of peritoneal lavage (ascites fluid) and blood (0.5 mL) were randomly collected pre-surgery and at 24 h (5 animals), 48 h (3 animals), 72 h (1 animal), and 96 h (2 animals) post treatment. The samples were sent randomly to the Microbiology Department of our hospital and were seeded during 48 h on blood and agar chocolate, MacConkey, and anaerobiosis culture plates.

### 4.7. Histology Assays

#### Histological Analysis

Samples of 5 mm^3^ were fixed in 10% formaldehyde at room temperature, embedded in paraffin and cut into 5-micron-thick slices in a Micron HM360 microtome.

Sections were stained with hematoxylin-eosin to identify and evaluate tissue damage and inflammatory cells infiltration.

An important point was to localize both types of hASCs cells (hASCs, hASCs-MOD,) in inflammatory tissues, so cells were stained with 1,1′-Dioctadecyl-3,3,3′,3′-tetramethylindocarbocyanine perchlorate (DiL, Invitrogen, CA, USA) before applications in three animals of different groups.

To assess the effect of hASCs cells, with and without modification, on abdominal sepsis after CLP, the following histological parameters by hematoxylin-eosin stain were studied:In the cecum, at the point of ligation, the number of neutrophils (cells characteristic of acute inflammation), plasma cells (chronic inflammation), M1 (CD68+) and M2 (CD163+) macrophages as innate immune cells were quantified.Quantitatively assessment was carried out by counting the number of neutrophils, plasma cells, and macrophages in 10 fields per high-powered field (HPF; 400×).In three sections of the intestinal loops (cecum, colon, and small intestine) a semi-quantitative assessment was performed to determinate the degree of tissue damage of their walls and thus observe the anti-inflammatory effect of hASC and hASC-MOD. It was evaluated according to the following scale: 0 = no damage, 1 = minimal damage, 2 = mild damage, 3 = moderate damage, and 4 = severe damage.The spleen, being the main blood-filtration organ during sepsis and susceptible to modification due to the immune response, was considered important for the evaluation of the treatment-related histology in our study. The compartments in the white pulp were evaluated, also considering the changes in size and cellularity according to Elmore SA et al.’s criteria (2006) (grading changes: 0 = normal, 1 = minimal, 2 = mild, 3 = moderate, and 4 = marked). The immunohistochemical studies were used in the detection of macrophages. The histology sections were deparaffinized and rehydrated before blocking endogenous peroxidase activity with H_2_O_2_ (0.3%) in methanol. Subsequently, the slides were rinsed with PBS and incubated with primary antibodies in a moist chamber at room temperature. The sections were then incubated with biotinylated anti-rabbit IgG and LBA (DAKO) for 25 min at room temperature, rinsed with PBS and immersed for 25 min in avidin peroxidase. The immunostaining reaction product was developed using diaminobenzidine and counterstaining was performed with hematoxylin. The specificity of the immunohistochemical procedure was confirmed by incubation of sections with non-immune serum instead of a primary antibody. The primary antibodies used were anti-CD68 antibody (Abcam ab-125212 antibody) and anti-CD163 antibody (Bioss bs-23127R antibody).

The histological slides were studied under a Zeiss Axiophot 2 microscope and photographed with an AxiocamHRc camera. The evaluation and quantification were performed by the same researcher who was unaware of the groupings. In all histological studies, for the quantification of these cells, twenty contiguous non-overlapping fields (400×) per slide from each group were counted.

### 4.8. Cytokine Multiplex Arrays Measurements

To assess changes in the secretion of inflammatory factors in blood and peritoneal fluid caused by the surgical procedure and treatments, Bio-Plex Pro™ mouse cytokine, chemokine, and growth factor magnetic bead-based assays (23-plex assay, Bio-Rad, Hercules, CA, USA) were conducted using a Luminex system (Milliplex MAP, Merck, Darmstadt, Germany). The assays were performed in triplicate for each sample, following the manufacturer’s instructions. Plasma samples obtained 5 and 11 days after CLP surgery were collected (200 µL each). The molecules were grouped according to their functional properties for analysis. The pro-inflammatory cytokines included IFN-γ, IL-12 (p40 and p70), IL-17, IL-1α, IL-1β, GM-CSF, IL-6, IL-2, and TNF-α. The anti-inflammatory cytokines comprised IL-4 and IL-10. Chemokines evaluated were KC (keratinocyte-derived chemokine), MIP-1a, MIP-1b, MCP-1, RANTES, and eotaxin. Other cytokines or growth factors analyzed included G-CSF, IL-3, IL-5, IL-9, and IL-13.

### 4.9. Statistical Analysis

This study applied a mixed-method design to quantify observations made on each variable. One-way analysis of variance (ANOVA) followed by Tukey’s post hoc comparisons tests were performed for statistical analysis. Data are presented as mean +/− standard deviation (SD). Statistical significance was considered at a *p*-value less than 0.05. The analyses were conducted using the SPSS statistical package, version 27.0, software for Windows (SPSS, Chicago, IL, USA).

## 5. Conclusions

This study successfully generated a reproducible mouse model of peritonitis. The findings demonstrate that the peritoneal administration of both wild-type (WT) and genetically modified adipose-derived mesenchymal stem cells (ASCs) expressing CXCR4 and IL-10 is an effective and safe therapy in this mouse peritonitis model. The positive outcomes suggest the potential for further investigations and, if successful, may pave the way for clinical application. Additional studies, particularly in larger animal models, would be beneficial to assess the viability and safety of transferring this therapeutic approach to clinical trials.

## Figures and Tables

**Figure 1 ijms-25-00520-f001:**
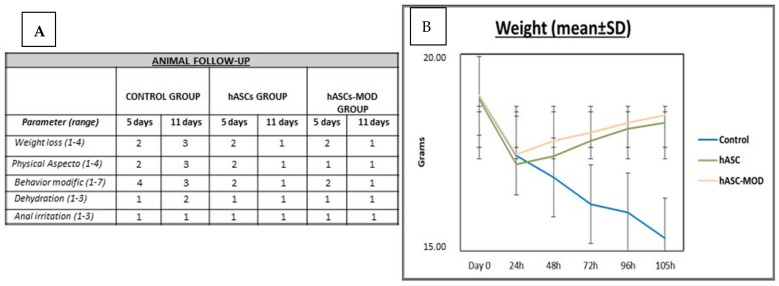
Follow-up of the animals. (**A**) Daily follow-up table. All columns include data at 5 and 11 days of follow-up. The minor value indicates better status in all parameters. (**B**) Weight evolution. (mean ± SD).

**Figure 2 ijms-25-00520-f002:**
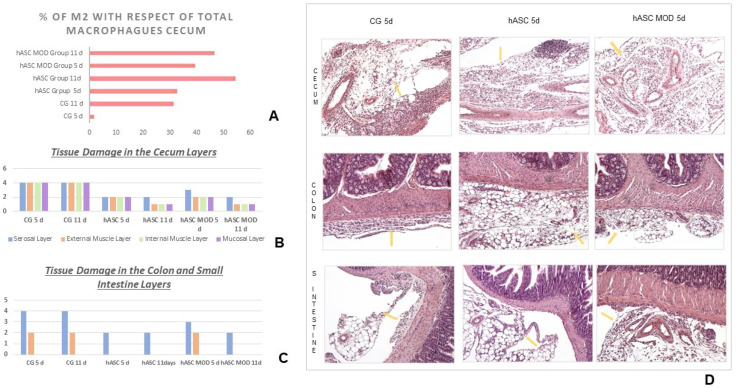
Tissue damage in the intestinal loops. (**A**) Macrophage population in the cecum. Percentage of macrophages M2(CD163+) with respect to the total number of macrophages(CD68+) found in the different study groups. (**B**) Tissue damage in the cecum layers observed in the three studied groups. (**C**) Tissue damage in the colon and small intestine layers. (**D**) Microphotographs of intestinal loops after 5 days of evolution. Hematoxylin-eosin, 400×. In the cecum of the control group, inflammatory infiltration is rather abundant, while it is milder in the cell-treated groups. In the colon and small intestine of the control group, all tissue layers are very damaged. On the other hand, in the hASCs and hASCs-MOD groups, the inflammatory infiltrate is reduced and only the outer layers (serosa and outer muscular) are affected. Yellow arrows: inflammatory infiltration.

**Figure 3 ijms-25-00520-f003:**
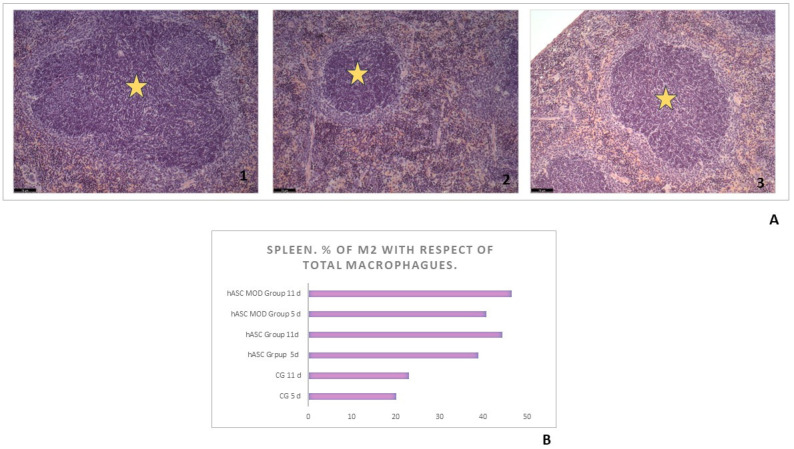
Spleen reaction. (**A**) Microphotographs of spleen at day 11, hematoxylin-eosin, 100×. (1) Control group. The area of periarterial lymphatic sheath (PALS) is significantly increased. (2) In the hASCs group, there is a minimal increase in PALS dimension. (3) In the hASCs-MOD group, there is is mild increase in the size of PALS. (**B**) Percentage of macrophages M2 (CD163+) with respect to total macrophages (CD68+) found in the spleen in the different study groups. Yellow asterisk: PALS zone.

**Table 1 ijms-25-00520-t001:** The mean results of cytokines (CTK) and chemokines at 5 days post-treatment are presented. Significantly elevated values are highlighted in bold, while significantly decreased values are highlighted in shading when compared to the sham group.

Cytokine and Chemokine Results at 5 Days (Mean)					
		SHAM	Control Group	hASC Group	hASC MOD Group
**CTK PRO-INFLAMMATORY**	**IL-1a**	7.7	11.46	9.6	9.17
	**IL-1b**	3.68	**14.92**	3.25	6.85
	**IL-2**	3.81	5.55	3.75	3.32
	**IL-12**	132.8	**191.63**	*45.3*	*76*
	**IL-17**	53	30.19	*12.31*	*9.67*
	**IFN-γ**	11.9	**34.45**	8.02	13
	**TNF-α**	27.12	**53.09**	21.53	28
**CTK ANTI-INFLAMMATORY**	**IL-4**	3.07	2.46	**6.79**	4.26
	**IL-10**	30.22	32.87	33.04	**43.8**
**CHEMOKINES**	**Eotaxin**	730.72	128.18	604.46	504.62
	**G-CSF**	46.7	60.75	49.98	57.84
	**GM-CSF**	18.18	**38.54**	17.21	*3.06*
	**KC**	16.33	**27.83**	12.52	15.96
	**MCP-1**	87.79	147.12	102.2	125.26
	**MIP-1a**	1.28	**2.43**	1.1	1.48
	**MIP-1b**	10.62	**21.82**	11.62	14.17
	**RANTES**	69.45	30.54	34.88	24.42
**CTK REGULATORY**	**IL-3**	0.76	**1.76**	0.45	0.71
	**IL-5**	4.49	**10.76**	4.8	**10.27**
	**IL-6**	1.16	**3.69**	2.83	2.63
	**IL-9**	10.06	**19.79**	*1.13*	6.63
	**IL-13**	17.86	**81.7**	23.11	39.73

CTK, cytokine; G-CSF, granulocyte colony-stimulating factor; GM-CSF, Granulocyte-macrophage colony-stimulating factor; hASC, human adipose-derived stem cells; IFN-γ, interferon gamma; IL, interleukin; KC, keratinocyte-derived chemokine; MCP-1, monocyte chemoat-tractant protein-1; MIP, macrophage inflammatory protein; MOD, modified; RANTES, regulated on activation, normal T cell expressed and secreted; TNF-α, tumor necrosis factor alpha.

## Data Availability

Data is contained within the article or Appendix A.

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
