# Peer review of "Study of the Effect of Wild-Type and Transiently Expressing CXCR4 and IL-10 Mesenchymal Stromal Cells in a Mouse Model of Peritonitis"

_ijms, 2023, doi:10.3390/ijms25010520_

Round 1

Reviewer 1 Report

Comments and Suggestions for Authors

This study shows the effects of treatment of hASC and hASC-MOD (expressing CXCR4 and IL10) on the immune system of the sepsis mouse models. The administration of ASCs might be effective in modifying the immune activities but the effects of expressing CXCR4 and IL10 seems not to be significant. 

1. In administrating the hASC-MOD, the expression of CXCR4 and IL10 in mice should be tested. IL10 expression seems to increase as seen in Tab.1. How about is CXCR4? The no-different effects of hASC-MOD versus hASC might be dependent on the non-expression in the mouse body.

2. In Tab.1, it’s hard to see bold and grey numbers. 

3. In results, the contents of the text seem to be different from the results of Supplement Tab.2.

For example, the number of neutrophiles in 4days hASC-MOD was increased according to supplement Tab.2, but the text said “In all study groups …, there was a decrease in this infiltrate,…(L160). There are several discrepancies of drawing in the text like this.

4. In Fig.2A and 3B, the dimension of the horizontal axis should be drawn.

5. In Fig.2D, the small marks indicating inflammatory infiltrate should be added to the pictures.

6. in Fig.3A, the small marks indicating T lymphocyte zones or the lymphocyte B dependent area should be drawn in the pictures. 

7. May “Figure 3A”s in L194 and L197 be “supplementary Tab.3”s?

Author Response

Dear Reviewer, 

We would like to thank you for your work on our manuscript “Study of the effect of wild-type and transiently expressing CXCR4 and IL10 mesenchymal stromal cells in a mouse model of peritonitis”. We feel that your comments have improved the article and made it more understandable to the reader.

We have tried to answer your questions and are at your disposal for any further clarification you may need in the future. In “Manuscript Rev” version, where we´ve incorporated the changes based on your comments.

Yours sincerely, 

Dra. García Gómez-Heras and Dr. García-Arranz.

Reviewer 2 Report

Comments and Suggestions for Authors

In the paper "Study of the effect of wild type and transiently expressing CXCR4 and IL10 mesenchymal stromal cells in a mouse model of peritonitis." The authors have demonstrated that the peritoneal administration of hASCs or genetically modified hASCs could be an efficient and safe therapy in mouse peritonitis. The manuscript needs to be thoroughly checked and revised for proper presentation of data.

1. Figure 1 can be divided into A and B and legends should be written explicitly.

2. Results headings should reflect the outcomes of the experiment.

For example. Follow-up can be written as "Assessment of subjects after surgery".

3. Microbiology can be written as Comparative assessment of microorganisms in various medium.

4. cytokines levels can be written as Perturbation in inflammatory markers to treatment conditions.

5. How molecular pathways are perturbed can be showed by a pathway diagram.

Author Response

(The authors gave the same response as above.)

Reviewer 3 Report

Comments and Suggestions for Authors

Review for the manuscript "Study of the effect of wild type and transiently expressing CXCR4 and IL10 mesenchymal stromal cells in a mouse model of peritonitis"

Overall comments: In this study, the authors intended to "a comparison of the safety and therapeutic effect of wild-type hASCs and hMSCs transiently expressing CXCR4 and IL-10 was made, aiming to generate an improved anti-inflammatory response for sepsis, in addition to standard antibiotic treatment". Although this is an interesting topic, I suggest some modifications before it can be accepted for publication.

TITLE: Please remove the period from the title.

ABSTRACT

            This section is adequate, however, I suggest modifying "Results (follow-up, microbiology, cytokine/chemokine levels and 29 mainly histopathology) showed that the infusion of both ASCs decreased the inflammatory state 30 and improved the resolution of the septic process compared to antibiotic treatment" for "Results showed that the infusion of both ASCs decreased the inflammatory state 30 and improved the resolution of the septic process compared to antibiotic treatment".

KEYWORDS

            There are too many keywords: Keywords: Mesenchymal stromal cells; sepsis; peritonitis; CXCR4; IL10; stem cell therapy; immunomodulation. Please, remove at least one.

INTRODUCTION

            I suggest not repeating the same references as in paragraphs 1-3. The authors can find these references in different databases such as PUBMED and EMBASE.

            In the lines 77-86 we can read "In the study, a single ASC dose of 1 million hASCs per animal 77 was used, in line with previous articles that have shown that MSCs appear to be effective. Furthermore, the antimicrobial activity of hASCs was evaluated by analyzing cytokines and chemokines levels in blood and peritoneal lavage, also by histology analyses of different mice tissues. Moreover, a study was conducted on the capacity of hASCs to reduce the systemic inflammatory state and organ dysfunction through the modulation of the immune response and expression of tissue protective/regenerative factors. Having said that, a comparison of the safety and therapeutic effect of wild-type hASCs and hMSCs transiently expressing CXCR4 and IL-10 was made, aiming to generate an improved anti-inflammatory response for sepsis, in addition to standard antibiotic treatment". What do you mean about "In the study"? Is this referring to the cited references (16-17) or in the author's study?
Moreover, some grammar and punctuation errors need to be corrected.

RESULTS or METHODS

            In line 86, the authors present the Results section; however, they present methodological procedures in 2.1 to 2.4 sections. After, we see some Results presentation (lines 139-141). After that, we again, see Methods in page 5-8.  In my opinion, the authors should clearly separate these sections.

            Furthermore, it is necessary to improve the quality of Table 1 and the images.

            On line 319, we see the Methods section again.

DISCUSSION

            This section is adequately performed.

REFERENCES

            As pointed out, I suggest removing repeated references in paragraphs 1-3 and including others.

Dear authors, I wish you good luck with this manuscript.

With regards

Comments on the Quality of English Language

Moderate changes are need.

Author Response

(The authors gave the same response as above.)

Round 2

Reviewer 1 Report

Comments and Suggestions for Authors

The authors have responded to all of my concerns.

Reviewer 3 Report

Comments and Suggestions for Authors

Dear authors,
Thank you for performing the modifications.

I wish you good luck with this manuscript.

With regards

Comments on the Quality of English Language

Minor